# Bladder Cancer: Uncovering the Predictive Role of NOTCH as an Emerging Candidate Biomarker for Therapeutic Strategies

**DOI:** 10.3390/cancers17183078

**Published:** 2025-09-20

**Authors:** Chiara Cusumano, Federica Squillante, Marco Roma, Roberto Miano, Maria Pia Felli

**Affiliations:** 1Unit of Urology, Department of Surgical Sciences, Policlinico Tor Vergata University Hospital, University of Tor Vergata, 00133 Rome, Italy; chiara.cusumano@ptvonline.it (C.C.); marco.roma@ptvonline.it (M.R.); mianor@virgilio.it (R.M.); 2Department of Experimental Medicine, Sapienza University of Rome, 00161 Rome, Italy; federica.squillante@uniroma1.it

**Keywords:** NOTCH pathway, muscle-invasive and non-muscle-invasive bladder cancer, NOTCH3, biomarkers

## Abstract

Molecular characterisation of bladder cancer (BCa) and particularly the classification of tumours into molecular subtypes, could facilitate the development of novel therapeutics for high-risk muscle-invasive bladder cancer (MIBC). This review will examine pathways implicated in MIBC, including growth factors and DNA–RNA modifying enzymes, with a focus on the differential roles of NOTCH receptors. In contrast to the tumour suppressor NOTCH1, NOTCH2 and NOTCH3 have demonstrated an oncogenic role in BCa and correlation with poor prognosis. Considering the crucial role of the NOTCH pathway in bladder cancer, we will discuss in our review the predictive value of the NOTCH pathway in BCa prognosis and its potential role in therapy.

## 1. Introduction

Bladder cancer (BCa) is one of the most common cancers worldwide. Men are affected much more often than women—globally, men have incidence and death rate of 9.5 and 3.3 per 100,000 people, which is about four times higher than the rates seen in women. The highest numbers of new cases are found in regions like southern and western Europe and North America [1]. A number of BCa risk factors were identified as follows: tobacco smoking (the most prevalent one [2]), occupational exposures [3,4], environmental exposures (such as X or gamma-radiation and arsenic), medications (cyclophosphamide, heavy consumption of phenacetin-containing analgesics, or pelvic RT), dietary factors, and Schistosoma haematobium infection [5].

## 2. Classification, Clinical Aspects, and Therapy

Most of these tumours originate from the urothelium, a stratified epithelial structure that lines the urine-exposed surface of the bladder. These urothelial tumours account for 75% of cases of non-muscle-invasive (NMIB) diseases confined to the mucosa, and muscle-invasive (MIB) diseases extended to deeper layers of the bladder wall or forming metastases in the remaining 25% of patients [6]. According to the TNM classification system, urothelial tumour confined to the mucosa are categorised as a stage Ta, while those invading the lamina propria are classified as a stage T1. High grade intraepithelial tumours limited to the mucosa are defined as a carcinoma in situ (CIS). CIS can be further categorised into three subtypes: primary CIS, which occurs in the absence of any prior or concurrent papillary tumours; secondary CIS, which arises during follow-up in patients previously diagnosed with non-CIS tumours; and concomitant CIS, which is identified in the presence of other urothelial tumours within the bladder [6].

Hematuria is the most significant symptom of BCa and the standard diagnostic methods include cystoscopy and the less invasive urine cytology, but with limited sensitivity in low-grade lesions [7]. Transurethral resection of the bladder tumour (TURBT) is the first therapeutic option for NMIBC; whereas, in patients with MIBC, the main option is the radical cistectomy (RC). TURBT can be supplemented with the bacillus Calmette–Guérin (BCG) intravesical instillations in selected patients, to prevent bladder cancer recurrence and progression [6].

Recently, Asimakopoulos et al. revised novel intravesical agents as a second-line treatment for BCG-unresponsive high-risk NMIBC. Nadofaragene firadenovec is hypothesised as a potential therapeutic advancement for difficult-to-treat diseases. Also, mutated IL-15 based immunostimulatory fusion protein compound (N-803), in combination with BCG, showed antitumour activity, but low efficacy in monotherapy. Future research could identify the right intravesical agent for each specific patient, according to safety, results, and cost-effectiveness [8].

Due to its recurrent nature, BCa needs ongoing disease monitoring and expensive therapies that include the above-mentioned surgery, chemotherapy, and radiotherapy.

According to the “WHO 2024/2022 upgraded classification”, BCa is divided in papillary urothelial neoplasm of low malignant potential (PUNLMP), non-invasive papillary carcinoma low grade (LG), and high grade (HG). This system is an upgrade of the WHO 1973 classification that expresses for the bladder cancer a Grade from G1 to G3 [9].

Accurate risk stratification is essential for guiding therapeutic choices and predicting disease progression and survival [9].

Over the past decades, several classification systems and prognostic models have been developed as follows: the EORTC 2006 risk tables, based on the WHO 1973 classification [10]; the more recent EAU NMIBC 2021 guidelines, which integrate both WHO 1973; and the WHO 2004/2016 grading systems, which have introduced a four-tiered risk model (Table 1) [9].

Several negative prognostic markers have been identified, including the presence of carcinoma in situ (CIS), tumour persistence after re-TURBT, early recurrence, age >70, and female sex. The prognosis of the patients with MIBC is more difficult for the heterogeneity of the disease [11,12,13]. For NMIBC, cancer-specific mortality (CSM) was highest for T1HG (19.52%), followed by Tis (15.56%), similar for T1LG and TaHG (10.88% and 9.23%, respectively), and lowest for TaLG (3.76%) [14].

Asimakopoulos and colleagues [15] correlated the depth of lamina propria invasion in T1a-c and the extension of the lamina propria invasion to T1-microinvasive (T1m) or T1- extensive (T1e) to predict recurrence-free survival (RFS) and progression-free survival (PFS). The 5-year RFS was 47.5% and the 5-year PFS was 75.9% with a significant difference between T1c and T1a and between T1e and T1m. Therefore, these interesting findings support its incorporation into innovative prognostic nomograms [15]. Primary MIBC is defined as the presentation of a muscle-invasive disease at initial diagnosis, while secondary MIBC presumes that non-muscle-invasive disease later progresses to MIBC [16,17,18]. Secondary MIBC is associated with poorer cancer-specific survival compared to primary MIBC, largely due to delayed radical cystectomy (RC), higher TURBT frequency, and reduced responsiveness to neoadjuvant chemotherapy (NAC) [19,20,21]. The 5- and 10-year overall survival (OS) rates for patients with progressive muscle-invasive bladder cancer (MIBC) were 37% and 20%, respectively. In contrast, patients with de novo MIBC exhibited significantly improved survival outcomes, with 5- and 10-year OS rates of 49% and 39%, respectively [22].

Therapeutic strategies for MIBC include RC, trimodal therapy (TMT)—that combine TURBT, chemotherapy, and radiotherapy—NAC, followed by RC, and bladder-sparing strategies. Evidence suggests comparable long-term oncological outcomes between TMT and RC, with TMT offering a better quality of life but at higher costs. The 10-year overall survival (OS), disease-specific survival (DSS), and recurrence-free survival (RFS) rates were 30.9% for TMT and 35.1% for RC. The mean 10-year DSS was 50.9% for TMT and 57.8% for RC. NAC improves pathological complete response (pT0) and survival, particularly when four cycles are administered and showed 5% absolute improvement in overall survival at 5 years [23,24,25].

Salvage RC (SV-RC), required in approximately 20% of patients undergoing bladder-sparing therapy, carries a high complication rate but remains a critical option in cases of failure or recurrence [26].

Cystectomy is associated with a 5–15% risk of pelvic recurrence, typically occurring within the first 24 months post-procedure. The median survival following diagnosis of muscle-invasive bladder cancer (MIBC) ranges from 4 to 8 months. Distant recurrence is observed in up to 50% of patients who undergo radical cystectomy (RC) for MIBC. Systemic recurrence is more prevalent in cases with locally advanced disease (pT3/4), with reported rates ranging from 32% to 62%, and in patients with lymph node (LN) involvement, where recurrence rates vary from 52% to 70%. For patients with progressive disease treated with platinum-based chemotherapy, median survival is generally between 9 and 26 months [27,28,29].

Considering the high impact of the disease, advances in molecular diagnostics can significantly modify the management of BCa, particularly the unveiling of biomarkers that facilitate early detection and provide prognostic information. Therefore, we will discuss past and very recent research in molecular markers to outline interesting advances, particularly the promising role of NOTCH signalling.

## 3. Biomarker in Bladder Cancer

Recently, more studies focused on the detection of molecular markers in urine, tissue, or blood in the early diagnosis of a bladder tumour. This entails the opportunity to diagnose the disease earlier, stratify the risk of the patient, improve prognosis, and facilitate more specific therapy. Biomarkers detected from bladder cancer samples will be able to detect patients who are at risk of recurrence or assess the tumour’s therapeutic response in advance. Another less-invasive option is a fluid biopsy by obtaining miRNA from serum. Urine biopsy is an emerging alternative in new biomarkers’ detection with the potential to provide insight into disease development, diagnosis, and prognosis with a minimal intervention for the patient [30].

Overall, NMIBC (stages Tis, Ta, and T1) and MIBC (stages T2–T4) show distinct molecular profiles and, within each disease category, they exhibit great molecular heterogeneity. Tumours classified as T1 often have molecular features that overlap with MIBC and diverge greatly from low-grade Ta tumours [31,32,33].

There is no obligatory pathway from NMIBC to MIBC, and it seems that their pathogenesis pathway does not overlap. From histopathological and molecular data, CIS seems to be the stage prior to MIBC, whereas the majority of papillary NMIBC originates from a urothelium with apparently normal features. However, NMIBC could progress to MIBC, especially in those patients with tumours invading the lamina propria [34].

Molecular characterisation of both NMIBC and MIB has received considerable attention and improved our knowledge for subtype classification, which may have a transformative potential in oncological care. We will summarise the most represented alterations as molecular biomarkers in BCa and consider the recent advances on and the emerging role of NOTCH3.

## 4. Molecular Mutations in NMIBC

Biomarkers are objectively measured and used as indicators to characterise disease processes or response to therapeutic interventions. In the heterogeneous BCa, analysis of the tissue detected multiple biomarkers including growth factors and growth factors’ receptors (FGFR3 and VEGF-C), transcription factors (GATA3, FOXA1 and P53), cell-free DNA, and epigenetic modification and non-coding RNA [35,36,37].

Growth factors: FGFR3 plays an important role in different cellular processes, including regulation of proliferation, differentiation, and apoptosis. Nevertheless, it is a major actor in angiogenesis and wound healing as well as embryogenesis [35,38]. This growth factor has been frequently detected in the lower stage of bladder cancer. In NMIBC, up to 15% had a somatic change in FGFR3, whereas 7% had FGFR1 amplification and 6% gene fusion [35,36,37,38,39]. FGFR3 mutations are considered early oncogenic drivers, associated with a favourable prognosis, lower risk of progression, and distinct molecular pathway compared to high-grade or MIBC. FGFR3 mutations also predict response to FGFR inhibitors, such as erdafitinib that is approved for advanced or metastatic BCa with FGFR3 alterations [38,40,41,42].

FGFR3 stimulates Stearoyl-CoA Desaturase 1 (SCD1) activity and thus promotes tumour growth in bladder tumour cells [30]. High levels of SCD mRNA and protein have been associated with poor prognosis in patients with bladder cancer. Its upregulation is restricted to cancer tissue samples, while absent in adjacent non-tumour tissue [43]. RAS mutations (HRAS, KRAS, and NRAS) are less frequent, approximately 10–20% of bladder tumours, and are not associated with tumour grade or stage. FGFR3 and RAS mutations are mutually exclusive oncogenic events that define distinct molecular subtypes and pathway of tumorigenesis. This suggests that either FGFR3 or RAS activation is sufficient for tumorigenesis in each tumour, but both are rarely found together [44,45,46].

DNA–RNA modifying enzymes: Growth factor mutations are frequent co-occurring lesions with those of either DNA mutators or RNA editors, according to the first preference identified for the nucleic acid substrate [47]. The most common mutations were those of FGFR3, then STAG2, and other chromatin regulators (KDM6A, KMT2D, KMD2C) and PIK3CA mutations [48]. In this regard, loss of STAG2 in NMIBC indicates a lower risk of progression to MIBC, thus confirming that STAG2 loss is a good prognostic factor in the context of progression free survival (PFS) [49]. KDM6A mutations are more common in stage Ta than in stage T1, whereas ARID1A mutations are more common in stage T1 tumours [48]. The APOBEC are a family of cytidine-deaminase that have been classified as DNA mutators or RNA editors [50]. Nearly 30% of patients with NMIBC displayed APOBEC target mutations in PIK3CA hotspot codons, often associated with mutations in FGFR3 or RAS genes [48], suggesting that both RAS–MAPK (mitogen-activated protein kinase) and PIK3 signalling are activated in the majority of NMIBC [51].

## 5. Molecular Mutations in MIBC and Metastatic Disease

MIBC exhibits remarkable intratumour genetic heterogeneity which is a considerable obstacle for both scientists and clinicians when approaching the development of candidate markers or of new therapeutic agents [52]. Heterogeneity occurs at multiple levels and directly affects clinical care.

Growth factors: Despite the fact that upregulated expression of FGFR3 is expected to be very common, MIBC showed less frequency of activating point mutations in FGFR3 and PIK3CA than in NMIBC [52]. In around 70% of MIBC, activation of the RAS–MAPK and PI3K pathways occurred, usually due to mutation or upregulation of upstream regulators. Gain-of-function mutations of ERBB2 and ERBB3, or amplification of ERBB2 and EGFR are also included [53].

Pathways implicated in MIBC include the hepatocyte growth factor receptor/cMET, a receptor tyrosine kinase (RTK), which upon ligand binding, activates different steps in the signal transduction cascade that rules cell survival, proliferation, and invasion. c-MET overexpression was found in several carcinomas, and its phosphorylated form is only detected in high pT stage and associated with malignant aggressiveness and poor survival of patients with BCa [54]. Identification of crosstalk partners of RTKs implicated in the development of bladder cancer may play an important role in predicting aggressiveness. To this end, Naguib and colleagues [48] have placed their attention in the expression status of c-MET and HER2 in 40 human BCa patients (study group) and 20 with benign disease (controls). c-MET and HER2 gene expression were found significantly higher in the study group compared to the controls, with c-MET and HER2 overexpression both associated with pathological stage, muscle invasion, and node involvement, whereas only c-MET overexpression is associated with tumour grade. In patients with expression of both c-MET and HER2, disease-free survival rates were lowest among all studied patients, showing the possibility of a correlation between c-MET and HER2 gene overexpression and unfavourable clinical outcomes in patients with BCa.

DNA–RNA modifying enzymes: Almost all MIBC exhibit a loss of cell cycle checkpoints through TP53, RB1, and/or ATM mutations and/or alterations of their regulators, such as E2F3 and MDM2 amplification, mutation of FBXW7, known to regulate NOTCH [55], and deletion of CDKN2A. DNA damage response and DNA repair pathways (for example, loss of function of ATM or ERCC2 mutation) [56] are also affected; ERCC2 is involved in 24% of T1 tumours [33]. Overall, the involvement of chromatin modifiers in MIBC is like that in NMIBC, but they differ for the mutation’s distribution.

The specific protein Sp1 was discovered to be another promising specific marker implicated in carcinogenesis and cancer progression. This belongs to the Sp family of Kruppel factors and binds specific promoter elements of its target genes, acting as a transcription factor (TF). Sp1 controls genes mainly responsible for many cellular processes: growth, apoptosis, or cycle regulation [57].

These genes appear to be involved in cellular functions that underlie tumour initiation and progression. In recent years, overexpression of the Sp1 gene has been found to be implicated in the development of various tumours, such as stomach cancer and pancreatic adenocarcinoma [58,59].

A study by Zhu et al. is the first systematic report that identified Sp1 gene expression as a prognostic marker in bladder urothelial cancer. Their investigation associated high-Sp1 expression with poor patient survival and with higher histological grades and in samples of patient with metastases [60]. Therefore, they identified Sp1 as a potential independent prognostic biomarker, especially for patients undergoing surgery for bladder tumour. It could be a potential tool to select patients with an aggressive type of tumour and, consequently, with poor clinical outcomes [30].

However, molecular characterisation of both non-muscle-invasive [32,61] and muscle-invasive bladder cancer and, above all, the classification of tumours into molecular subtypes provided a promising basis for the development of new therapies in MIBC [62].

Multiple active clinical trials (ClinicalTrials.gov) deals with growth factors in MIBC. FGFR and MET are two of the key growth factors driving bladder cancer development and progression. Their central role in tumour biology has made them prime candidates for therapeutic targeting, which is why ongoing clinical trials are focusing on evaluating inhibitors against these pathways (Table 2).

Later in 2014, three interesting papers focused on the role of the NOTCH pathway in bladder cancer, as reported in [63].

## 6. The NOTCH Pathway as a Biomarker and Therapeutic Target

Mammalian NOTCH receptors are four single-pass transmembrane proteins (NOTCH1–2–3–4) that play a crucial role in regulating cell differentiation, proliferation, and invasion. It is a highly conserved cell-to-cell communication system. Upon NOTCH ligand, DLL (DLL1–4) and JAG (JAG1–2), interaction, the receptor undergoes a conformational change to expose sites S2 and S3 for proteolysis (Figure 1). This enables two sequential proteolytic cleavages of the receptor, first by a TNF-alpha converting enzyme (ADAM17, a membrane metalloprotease) and then by gamma-secretase (GS) [64]. In the canonical NOTCH pathway (Figure 1), the active NOTCH-intracellular domain (NICD) translocates into the nucleus where it interacts with CSL TF to modulate expression of target genes [65]. In contrast to many other pathways, NOTCH signalling can be either oncogenic- or tumour-suppressive, depending on the tissue type and cell context [66].

NOTCH have been extensively studied in hematopoietic tumours, and two seminal papers established the role of NOTCH1 in acute T-cell leukaemia [67,68]. More recently, growing evidence associated overexpression and activating mutations of NOTCH3 to T-cell leukaemia development [50] and progression [69]. In various solid tumours, oncogenic NOTCH supports malignant progression. Hyperactive NOTCH signalling is implicated in tumour growth and metastasis [70], thus frequently leading to poor prognosis in melanoma [71], non-small-cell lung cancer [72], and colorectal cancer [55] patients.

In small-cell lung carcinoma [73] and acute myeloid leukaemia [74], in contrast, NOTCH inactivation was described. Although the dissimilar role of the different NOTCH receptors needs to be considered in therapeutic design, the NOTCH pathway represents a compelling target for new drug development [65,75,76,77].

Until today, little has been known about NOTCH in bladder cancer. According to the literature, the situation is rather inconclusive, studies have proposed both tumour-suppressive and oncogenic functions for NOTCH signalling [63,78,79,80,81]. Starting from the suppressive role of NOTCH1, we will further extend our discussion to other NOTCH receptors (NOTCH2–3 and 4), focusing on the newly reported and promising data on NOTCH3.

Based on the distinct clinical characteristics and manifestations of NMIBC and MIBC, we interrogated the cBioPortal data set about the percentages of NOTCH1,2,3, and 4 receptor mutations in the two types of bladder cancer (NMIBC, MIBC) and in overall bladder cancer (Overall BC) cases, as shown in Table 3. Of note, our analysis highlights a higher percentage for NOTCH3 in MIBC than in NMIBC, thus suggesting its mutation as a critical element in the pathogenesis of the disease.

## 7. NOTCH1

In 2014, Greife and colleagues [82] examined the canonical NOTCH1 signalling pathway and found it to be downregulated, along with its ligand DLL1, throughout all stages and histological grades of bladder cancer, with the lowest levels observed in high-stage tumours. Consistently, canonical NOTCH signalling activity—assessed in a panel of bladder cancer cell lines exhibiting distinct invasive capabilities and not restricted to any specific NOTCH receptor—was particularly suppressed in invasive cell lines compared to non-invasive ones.

In the same year, Rampias et al. [80] investigated the involvement of the NOTCH signalling pathway in bladder cancer and reported that the loss of a NOTCH1 gene copy number was commonly observed in a cohort of 72 patients, which included both non-muscle-invasive and muscle-invasive cases. No copy number alterations were noted for NOTCH2 or NOTCH3. Ultimately, their findings supported a tumour-suppressive function for NOTCH1, and they proposed a similar suppressive role for NOTCH2 and NOTCH3.

Later, Maraver and colleagues [83] confirmed the tumour suppressive role of NOTCH1, but they observed another role for NOTCH2, whereas NOTCH3 was not studied. They found that the NOTCH pathway maintains the epithelial phenotype in bladder cells via its effector HES1-transcriptional target of the NOTCH pathway. Loss of NOTCH activity leads to the development of more aggressive, mesenchymal, and invasive tumours, by upregulating key mediators and effectors of the epithelial–mesenchymal transition (EMT), including SNAIL, SLUG, ZEB2, and VIMENTIN, and downregulating the epithelial marker E-CADHERIN [63,83,84]. In agreement with the idea that the NOTCH signalling is tumour-suppressive in squamous histology cancers and since the urinary bladder is a stratified epithelium with the potential to develop into squamous cell carcinoma, Maraver plausibly hypothesised a tumour-suppressive role for the NOTCH pathway in this context. Their research of NOTCH mutations —based on combining data from murine genetic models, in vitro cellular assays, and human bladder cancer specimens—supports a tumour-suppressive role for the canonical NOTCH signalling pathway in BCa. As in other types of cancers, the NOTCH pathway loss of function in the bladder is associated with squamous cell carcinomas (SCCs). These studies reflect the dual behaviour of the NOTCH pathway, depending on the cellular type [63,83].

Both Rampias et al. [80] and Maraver et al. [83] developed transgenic mouse models with bladder-specific deletion of two components of the γ-secretase complex—Psen (PsenKO) and Nicastrin (NicastrinKO) blocking NOTCH signalling (Figure 1). These genetic alterations led to enhanced bladder tumour formation, occurring either spontaneously [80] or following exposure to a chemical carcinogen [85]. The resulting tumours exhibited phosphorylation of ERK1 and/or ERK2, along with increased expression of the proliferation markers Ki67 and cyclin D1. All tumours were characterised by the absence of nuclear N1ICD and the presence of basal cell markers TP63 and cytokeratin 5 (KRT5), indicating a basal-like phenotype and suggesting an origin from the basal or intermediate urothelial layers, thus suggesting that inactive NOTCH1 is related to tumour development [82].

On the other hand, the clinical significance and targetability of NOTCH and MAPK signalling for bladder cancer was suggested by Schulz and colleagues [86]. They evaluated the activity of NOTCH1 and MAPK pathways in advanced stages (III and IV) of bladder cancer tissue samples from patients treated with radical cystectomy and assessed their relationship with clinical outcomes, such as cancer-specific survival and overall survival. They observed on a murine xenograft model of human BCa cells that NOTCH1 and MAPK signalling marked two distinct tumour cell subpopulations, and both contribute to tumour progression.

Interestingly, the combined activity of both pathways (NOTCH1 and MAPK) was associated with poorer cancer-specific and overall survival, acting as an autonomous prognostic factor. They hypothesised that NOTCH1 in BCa may induce the expression of dual-specificity phosphatases (DUSPs), leading to repression of MAPK downstream signalling molecules.

These pathways can be repressed with the gamma-secretase inhibitor (GSI) dibenzazepine (DBZ, Axon Medchem) and the MEK inhibitor selumetinib (AZD), respectively [86]. The combinatorial inhibition of NOTCH and MAPK signalling strongly suppressed tumour growth. These results proposed a new concept for BCa therapy, which advocates specific and simultaneous targeting of both NOTCH and MAPK signalling [86].

## 8. NOTCH2

In 2016, Hayashi and colleagues investigated the role of NOTCH2 in bladder cancer, especially regarding its involvement in epithelial–mesenchymal transition (EMT). They proposed NOTCH2 as an oncogene driving BCa progression [78]. High NOTCH2 mRNA levels in MIBC were associated with more aggressive disease features and poorer patient prognosis, based on their own clinical samples and analysis of The Cancer Genome Atlas (TCGA) data set.

In their orthotopic xenografts, stable silencing of NOTCH2 significantly inhibited tumour growth, reducing NOTCH2 and HEY1 expression and decreasing EMT and stem cell markers [78]. They tested, in vivo and in vitro, an antibody targeting the negative regulatory region (NRR) to stabilise the off conformations of NOTCH2. This NOTCH2 inactivating antibody NRR2Mab, which exhibits minimal cross-reactivity with other NOTCH receptors, was shown to suppress the expression of HEY1, as well as markers associated with mesenchymal traits and stem cells, relative to control IgG in BCa, without any toxic effects. NRR2Mab provided preclinical validation for the potential of NOTCH2 inhibition as a new therapeutic strategy for muscle-invasive bladder cancer [78].

Recently, Lin H. et al. interestingly demonstrated in bladder cancer that low-density lipoprotein receptor-related protein 1 (LRP1) has a potential oncogene function and it is associated with malignancy and immune evasion. LRP1, by inhibiting DLL4 ubiquitination, activates NOTCH2, thus triggering the progression of EMT and high C-C motif ligand 2 (CCL2) expression to induce M2-like macrophage polarisation. The authors further suggest the role of LRP1/DLL4/NOTCH2 as a mechanism of resistance to immunotherapy with immune-checkpoint inhibitors anti-PD1 and anti-PDL1 [87].

Interestingly, two recent manuscripts evidenced that NOTCH2 gene alterations correlate with immune infiltration and response to therapy in bladder cancer [88,89]. Indeed, the study by Nagumo Y et al. [89], by integrating gene alterations and tumour infiltrating lymphocyte profiling, could associate NOTCH2 mutations to clinical complete response to therapy (atezolizumab and radiation therapy) in MIBC. On the contrary, FGFR3 alterations are associated with non-complete response to immune-radiation therapy, in line with its association with aggressive BCa. The different therapy response was related to the CD8:Foxp3 lymphocyte ratio, higher in complete response to therapy, thus interestingly correlating NOTCH2 to the composition of the tumour microenvironment. The Authors hypothesise that NOTCH2, in association with CDK12, GNAS, AR1D1A gene alterations, could identify good responders to therapy in MIBC. The manuscript by Si-yu Chen et al. [88] supports the notion that NOTCH2 behaves as an oncogene. Besides the role of NOTCH2 in promoting cell proliferation and metastasis through EMT, this receptor is involved in BCa stemness, a crucial step in disease progression.

Further in vitro data correlates DLL4 protein not only with NOTCH2 but also with NOTCH3 receptor, as analysed in different bladder cancer cell lines. This supports the oncogenic role of the NOTCH2/NOTCH3/DLL4 axis in bladder cancer [90].

## 9. NOTCH3

Although influential studies by Rampias [80] and Maraver [83] reported NOTCH1 and NOTCH2 mutations in bladder cancer, NOTCH3 was not studied [63]. For that reason, the publicly available data set cBioPortal was interrogated for recurrent NOTCH3 receptor mutations (Figure 2).

Based on a large cohort of patients, derived from the integration of four independent genome-wide studies in bladder cancer, our analysis (Figure 2) distinguishes NOTCH3 mutations in non-muscle-invasive bladder cancer [61] from muscle-invasive cases [53,62,91,92].

Multiple driver mutations of NOTCH3 are associated with the more aggressive MIBC, whereas a NOTCH3 mutation has been detected in one NMIBC patient. Newly, based on publicly available data sets from TCGA, we performed a predictive analysis of NOTCH3 alterations with Polyphen-2 and SIFT predictions as shown in Table 4, supporting its oncogenic potential role.

One of the first evidence of NOTCH3 involvement in bladder cancer was reported by Zhang et al. in 2017 [93]. By analysing 59 urothelial cancer patients, they observed a higher NOTCH3 expression in human urothelial cancer tissues than in non-tumoral bladder tissue samples. Indeed, NOTCH3 knockdown decreased urothelial cancer cell proliferation in vitro and decreased xenograft tumour growth in vivo. Furthermore, they correlated high NOTCH3 expression to poor prognosis and short overall survival in BCa patients.

In this study, they also found that after NOTCH3 knockdown, urothelial cancer cells were more sensitive to cisplatin treatment, leading to almost complete elimination of cisplatin-resistant colonies, thus corroborating the observation that NOTCH3 blockade may prevent drug resistance in urothelial cancer cells.

Histone modifications are important mechanisms in regulating NOTCH3 expression/activity, as we previously demonstrated in tumoral context [94].

Increased NOTCH3 protein acetylation levels by suberoylanilide hydroxamic acid (SAHA), a histone deacetylase (HDAC) inhibitor, decreases NOTCH3 protein expression resulting in reduced tumour growth and enhanced sensitisation of urothelial cancer cells to cisplatinum. Considering high-NOTCH3 expression and poor prognosis demonstrated in urothelial carcinoma, HDAC inhibitors can become an effective therapeutic strategy for BCa [93].

More recently, Petrovic et al. [95] evaluated the immunohistochemical (IHC) expression of NOTCH3 in BCa and correlated NOTCH3 expression with histopathological and clinical parameters. More NOTCH3 positivity was observed in high-grade tumours and was associated with a higher risk of mortality, thus indicating NOTCH3 as a potential reliable IHC marker for selecting BCa patients who may require more intensive follow-up. A limit of the study was that NOTCH3 expression was not a statistically significant predictor of recurrence-free survival. Understanding the importance of the pathway and of the interplay between NOTCH receptors in bladder cancer could bring new possibilities for better controlling one of the most prevalent cancers in urology [95].

Despite the great progress in diagnosis and treatment of BCa over the past few decades, some patients have experienced relapse or even progression after treatment. Bin Y. et al. [96] identified ten NOTCH signalling pathway-related genes (SNW1, NOTCH3, ADAM17, MAML2, NUMBL, DTX2, DTX4, DTX3L, HES1, CIR1) and, based on their expression, classified bladder cancer into two subtypes C1 (upregulated ADAM17, DTX3L, MAML2, SNW1, NOTCH3, NUMBL) and C2. Afterwards, using a Lasso-Cox regression analysis, these ten genes were closely related to the prognosis of bladder cancer, Group C1 (high-risk) and Group C2 (low-risk). Additionally, by comparing the difference in common cell and immune-related scores of tumour microenvironment (TME), they found that Group C1 had stronger immune escape and a weaker ability to respond to immunotherapy. Consequently, they proposed a NOTCH signalling pathway-based bladder cancer classification and, in the context of immune infiltration, the potential application of genes of the NOTCH pathway as biomarkers for risk prediction and to improve clinical decision-making. Validation will require further in vivo and in vitro experiments.

More recently, Liu et al. [97] selectively studied NOTCH3 and found out that NOTCH3 upregulation in bladder cancer is closely associated with poor prognosis. These results come from analysis of NOTCH3 expression in BCa cohorts from the GEO and the TCGA database. To support their hypothesis, through ex vivo analysis, they demonstrated the upregulation of NOTCH3 and of its active form NICD, further confirmed in vitro by using human urothelial cells. Moreover, they showed in a xenograft model that depletion of NOTCH3 significantly reduced tumour volume and weight and tumour metastasis. To explain this effect, the authors [97] proposed a molecular mechanism in which NOTCH3 promotes the transcription of SPP1 (Osteopontin) by binding to the CSL elements in the promoter. Subsequently, SPP1 activates the PI3K-AKT axis to promote BCa cell proliferation, migration, and invasion, thus suggesting NOTCH3 targeting as a therapeutic approach.

These results are in line with the previous report by Nedjadi et al. [98], demonstrating that SPP1 is differentially expressed in the early stages of BCa and associated with poor prognosis of BCa patients. Although alterations in SPP1 gene have been reported but poorly represented in bladder cancer (2.19% of 411 bladder cases [98], we can hypothesise that NOTCH3-triggered molecular mechanism may be the prevailing one.

Of note, NOTCH3 variants have been described also in a case of primary bladder mucinous adenocarcinoma (PBA), a rare subtype of BCa accounting for less than 2% of all bladder malignancies [99].

## 10. NOTCH 4

There are no reports on NOTCH4 signalling in bladder cancer until 2016, when Hu J. et al., with protein profiling of BCa, reported NOTCH4 as a molecular marker to classify MIBC into two distinct groups. Group 1 was enriched with activation of the NOTCH4 pathway, which had poorer survival than Group 2 [100].

In 2018, a study by Goriki et al., although reporting about RNA expression and mutations analysis from the TCGA data set, could not identify NOTCH4 expression in BCa patients’ tissue or cell lines (RT-PCR or immunohistochemistry). Furthermore, in a TCGA data set analysis, they observed that the rate of genomic alterations of NOTCH4 (45.8%) is like that of the other NOTCH receptors, with copy number gain (26.7%) and mutations identified in 1.0% of patients [63].

For the first time, by multiomics data, NOTCH4 mutations were associated with response to immune checkpoint inhibitor (ICI) therapy. They found that NOTCH4 mutation is significantly associated with strengthened tumour immunogenicity, the expression of costimulatory molecules and activation of the antigen-processing machinery leading to a better response. NOTCH4 mutation positively correlates with infiltration of diverse immune cells (tumour-infiltrating lymphocytes), the release of immunostimulatory chemokines like CXCL10 and CXCL9, and the recruitment of CD8 T cells and dendritic cells. Therefore, the inclusion of the NOTCH4 mutation as a biomarker in a perspective basket trial is worth being proposed. A limitation to the retrospective analysis may rely on patients receiving combinational therapy based on anti-PD1 and anti-PDL1 antibodies as well as the unravelling of the potential molecular mechanism by which mutated NOTCH4 sensitises patients to ICI therapy [101].

Like NOTCH1 and NOTCH2, also NOTCH4 has been proposed as a common modulator of EMT mechanism in urological tumours [102].

Notably, mutations in six genes, including NOTCH4, were detected in the circulating tumour DNA (ctDNA) of NMIBC, which disappeared after an intravenous infusion with ACT, consisting of dendritic cell mixed with cytokines induced killer cells. The report consists of data only from a single patient [103].

Although recent data increased our knowledge about NOTCH 4, new efforts are still required to better explore the role of this receptor in bladder cancer.

## 11. NOTCH-Combined Therapy in Bladder Cancer

NOTCH inhibition mostly relies on the use of small molecules, such as gammasecretase inhibitors (GSIs), that prevent the third cleavage necessary to activate NOTCH signalling (Figure 1), essentially approached in vitro [79,104,105] and in vivo experiments [86,106]. Although GSIs were included in clinical trials, they demonstrated limited applicability due to their adverse effects (gastrointestinal toxicity, atopic dermatitis, and skin cancer), associated with the ubiquitous activity of both NOTCH and GSIs [107].

In addition to NOTCH signalling, the specific enzymes also involved in NOTCH processing could be amenable targets to small molecule inhibition; however, they share lack of specificity as gammasecretase inhibition [108].

Therefore, the specific targeting of an individual NOTCH receptor with a monoclonal antibody is in development. Genentech fine-tuned antibodies directed against the specific NRR domain (anti-NRR NOTCH) to block any conformational change necessary to allow activating cleavage of the receptors. The efficacy of the NOTCH2 inactivating antibody NRR2Mab has been demonstrated in preclinical model of bladder cancer [78].

A novel frontier is the use of a dual-antibody, targeting both NOTCH2 and NOTCH3 (tarextumab, OMP-59R5), used in early phase clinical trials for gastrointestinal tumours, although with some limitation to success, due to bowel toxicity [109].

In constant development are new approaches to target NOTCH; indeed, a novel anti-NOTCH3 Antibody-Drug Conjugate (PF-06650808) exhibited a promising antitumour activity in a clinical study (phase 1 dose escalation) of breast cancer and in other solid tumours, thus suggesting its potential activity also in bladder cancer (NCT02129205) [110].

SERCA pump (sarco (endo) plasmic reticulum Ca^2+^ ATPase) inhibitors are an emerging therapeutic strategy in solid and haematological tumours [111]. The platinum-based compound is widely used in clinical applications, and it is an FDA-approved drug for combined treatment of bladder cancer [112]. Recently, a SERCA inhibitor (CAD204520), suppressing mutated NOTCH1 signalling, has been used in preclinical studies of an orthotopic T-cell leukaemia model and it is a well-tolerated drug [113]. Ongoing studies are analysing the effect of this drug also on other mutated NOTCH receptors. Thus, considering the high-rate mutation of NOTCH3 in BCa, this SERCA inhibitor could be considered in preclinical bladder cancer therapy studies.

Recently, an FDA-approved topical antifungal agent, Ciclopirox (CPX), that mechanistically suppressed NOTCH signalling activation, demonstrated a preclinical anticancer activity in a high-grade urothelial cancer model. Poor oral bioavailability and gastrointestinal toxicity were observed. The pharmacologic activity of CPX is under evaluation in the first-in-human Phase 1 trial (NCT03348514), whereas it has already been evaluated in a Phase 1 expansion cohort study of MIBC patients scheduled for RC (NCT04608045), as well as a Phase 2 trial of newly diagnosed and recurrent BCa patients scheduled for TURBT (NCT04525131) [114].

The finding was that NOTCH 3 mutations are frequent in MIBC (Figure 2) and, as described above, associated with poor prognosis and short overall survival, supporting the hypothesis to target this receptor in BCa therapeutic strategies. Indeed, silencing of NOTCH3 in two bladder cancer cell lines sensitised one cell line to cisplatin, which is known to inhibit the SERCA pump, just like the recently discovered CAD204520 that can block oncogenic NOTCH [93,113]. This could favour a future combined use of these drugs. Moreover, we interrogated the publicly available data sets (TCGA) to discover co-occurring mutational events of the above-mentioned Growth factors and modifiers with NOTCH 3 (Figure 3).

Our analysis showed that NOTCH3 mutations are highly represented, but, interestingly, the most frequent co-lesions with NOTCH 3 involves PIK3CA (*n* = 8 patients) or ERBB (*n* = 7 patients), both associated with high-risk BCa. Although there are no PIK3CA-targeted drugs specifically approved for bladder cancer, different PI3K inhibitors have been approved by FDA, but not in monotherapy for solid tumours, including bladder cancer [115].

This potential co-targeting could benefit from future in vitro and in vivo studies testing the efficacy.

According to our analysis, the NOTCH 3 mutation is co-occurring with the FGFR3 mutation, highly represented in MIBC and for which erdafitinib is the FDA-approved molecule [116], a drug included in the clinical trials reported in Table 2.

In contrast, c-MET resulted as a co-lesion with NOTCH 3 and FBXW7 in one of the patients of our analysis, thus suggesting the potential use of Crizotinib, a multi-target tyrosine kinase inhibitor (TKI) in bladder cancer [117] in this mutational setting.

Interestingly, ERBB2 is a co-lesion frequently and recurrently associated with NOTCH 3 in different mutational settings of MIBC, although each represented by very few patients (Figure 3). Currently, there is only intravesical administration of an antibody-drug conjugate (ADCs), such as Disitamab Vedotin targeting HER2, that has shown promising effects in the treatment of solid tumours, including NMIBC [118,119]. Therefore, these findings require new preclinical studies to suggest their future application for clinical trials.

Although there has been increased interest in NOTCH, there are no active clinical trials testing NOTCH pathway inhibitors—either as individual drugs or in combination—in patients with bladder cancer. The available evidence is limited to promising preclinical studies, in NOTCH2, a potential therapeutic target [78]. The identification of NOTCH3 gene co-lesions (Figure 3) and preclinical data on NOTCH2 may suggest the inclusion of these receptors in future studies.

Of note, the recent study by Hao N. et al. [120] observed that laminin, a major component of extracellular matrix, was significantly upregulated in patients with MIBC. They proposed a novel signalling pathway in which laminin promotes tumour cell proliferation and migration via the integrin α6β4/TRB3/JAG1/NOTCH axis. Subsequently, they proved that SAHM1, a peptide mimetic of a dominant negative form of MAML1, inhibits canonical NOTCH transcription complex formation. Combined treatment with chemotherapeutic HCPT and SAHM1 enhanced tumour growth inhibition and prolonged the overall survival of tumour-bearing mice, thus suggesting an innovative strategy for the clinical treatment of bladder cancer [120].

## 12. Conclusions and Future Perspectives

The dual role of the NOTCH pathway, suppressive or oncogenic in solid tumours, depends on the tissue type and cellular context, and may also rely on structural differences among the receptors. Given this complexity, further studies are warranted to dissect the individual NOTCH pathways in bladder cancer. Besides the disparate roles of NOTCH1 and NOTCH2 in bladder cancer, very recent works highlighted the oncogenic potential of NOTCH3 as a candidate biomarker for prognosis and for novel therapies in bladder cancer. Future in vitro and in vivo studies could expand our knowledge about NOTCH3-triggered molecular mechanisms in BCa, especially in MIBC. Based on what has been reported so far, ex vivo studies in BCa patients could suggest integrating NOTCH3 mutations analysis in the routine molecular profiling for high-risk BC. Regarding the few studies reported about NOTCH4, much effort needs to be spent on the role of this receptor in bladder cancer. GSIs are commonly referred to as NOTCH inhibitors, but, considering receptor structural differences, precise inhibition of a NOTCH receptor appears to be a more rational therapeutic strategy. The improvement of more specific drugs targeting single NOTCH receptor is advocated. Since multi-omics and artificial intelligence (AI) approaches have already proven effective in dissecting the bladder cancer microenvironment and identifying predictive biomarkers, these technologies could likewise be applied in the future to investigate the role of NOTCH and guide the development of tailored therapeutic strategies.

## Figures and Tables

**Figure 1 cancers-17-03078-f001:**
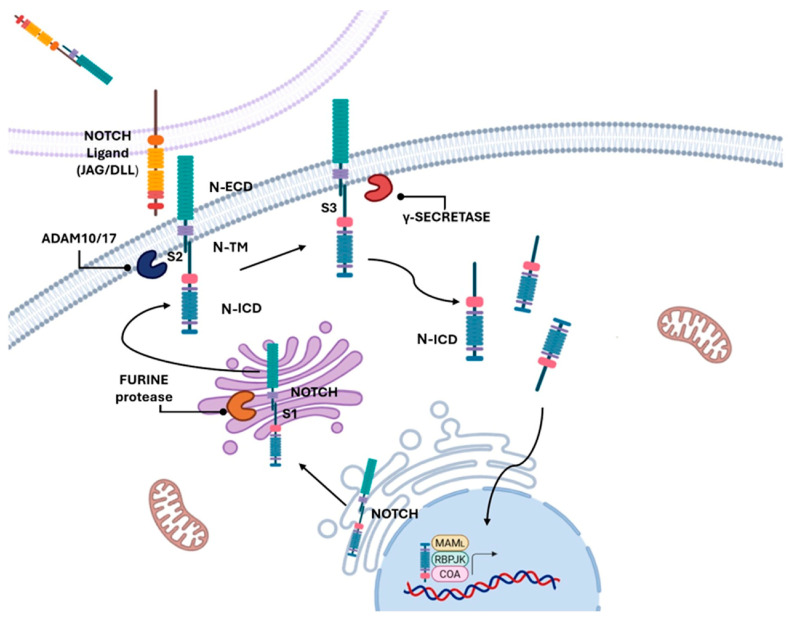
Schematic representation of the NOTCH activation pathway. In the Golgi apparatus, the receptor undergoes an S1 cleavage by a furin-like convertase, producing two non-covalently-associated subunits (N-ECD and N-ICD) that are transported to the plasma membrane as a functional heterodimer. Ligand binding (Delta, DLL1–4; Jagged/Serrate, Jag1–2) on a neighbouring cell induces a conformational change that exposes the S2 cleavage site, processed by ADAM10/17 metalloproteases. This is followed by the intramembrane S3 cleavage mediated by the γ-secretase complex, releasing the NOTCH intracellular domain (N-ICD). N-ICD translocates to the nucleus, where it interacts with the transcription factor CSL/RBP-Jκ and co-activators such as Mastermind, driving the transcription of target genes (Created with BioRender.com, accessed on 16 June 2025).

**Figure 2 cancers-17-03078-f002:**
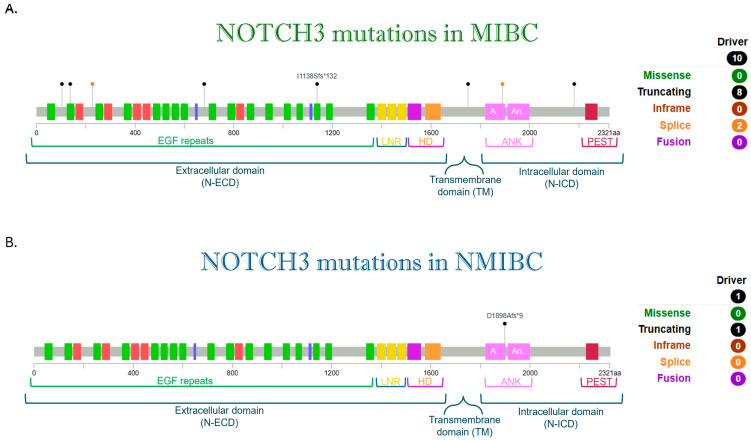
Map of mutations found in NOTCH3 receptor. The plot indicates mutations found in four independent genome-wide studies in (**A**) muscle-invasive bladder cancer [53,62,91,92] (*n* = 1203 patients) and in one independent genome-wide study in (**B**) non-muscle-invasive bladder cancer [61] (*n*= 105 patients) from cBioPortal (https://www.cbioportal.org/, accessed on 17 June 2025). We have reported only Driver mutations in MIBC (*n* = 8 patients) and NMIBC (*n* = 1 patient). Nonsense mutations (truncating) are indicated in black, and splice mutations are indicated in orange. The NOTCH protein consists of an extracellular region (*N*-ECD) featuring multiple epidermal growth factor (EGF)-like repeats, along with three cysteine-rich LIN12 and NOTCH repeats (LNR) and a heterodimerization domain (HD). This is followed by a hydrophobic segment at the carboxy-terminal end. The intracellular portion (N-ICD) contains several conserved domains, including ankyrin (ANK) repeats, nuclear localization signals (NLSs), and a PEST (proline–glutamate–serine–threonine) domain, involved in protein stability and degradation. Between the extracellular domain and the intracellular one there is the transmembrane domain (TM).

**Figure 3 cancers-17-03078-f003:**
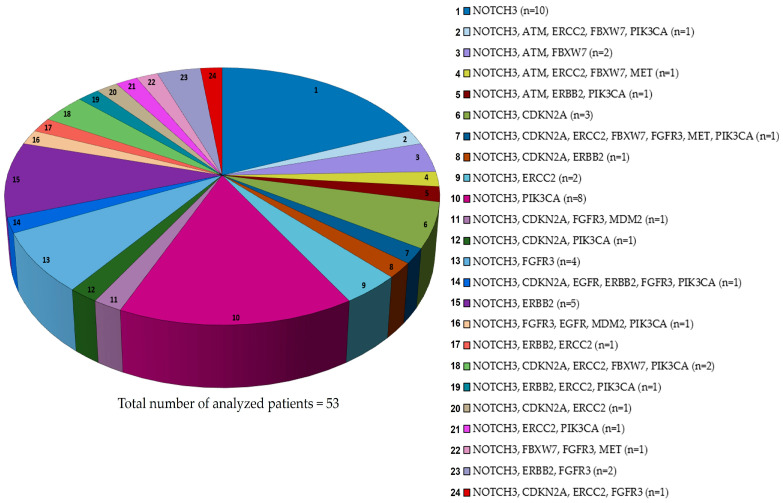
Distribution of NOTCH3 co-mutations with growth factor genes, DNA repair enzymes and cell cycle control genes. This pie chart illustrates the distribution of patient cases harbouring mutations in the NOTCH3 gene alongside alterations in other genes. The data were collected from four independent genome-wide studies in muscle-invasive bladder cancer (MIBC) [53,62,91,92], available on the cBioPortal platform (https://www.cbioportal.org/, accessed on 18 June 2025). Each coloured segment of the chart represents a distinct group of patients sharing a specific co-mutation profile between NOTCH3 and a particular gene. The total number of patients analysed is 53. *n* = analysed patients for each mutational status. The relative size of each slice corresponds to the number of patients identified with that specific combination, highlighting the heterogeneity and co-occurrence patterns of these genetic alterations.

**Table 1 cancers-17-03078-t001:** European Organisation for Research and Treatment of Cancer (EORTC), European Association of Urology Non-Muscle-Invasive Bladder Cancer (EAU NMIBC), carcinoma in situ (CIS), Bacillus Calmette–Guérin (BCG).

	EORTC 2006 Risk Tables [10]	EAU NMIBC Guidelines 2021 [9]
RISK STRATIFICATION	Classified NMIBC into three risk groups: low, intermediate and high-risk	Categorise NMIBC into low, intermediate, high and very high-risk groups
RISK FACTORS CONSIDERED	Tumour size, prior recurrence rate, tumour stage (Ta/T1), presence of CIS and grade, number of tumours	Similar parameters with the addition of patient age, tumour size ( > 3 cm), multiplicity and other factors
SCORING SYSTEM	Estimate the probability of recurrence and progression at 1 year and 5 years	Referenced but criticised for not estimating the risk of progression
LIMITATIONS	Molecular markers are not considered; Overestimates the risk of progression, especially in patients undergoing intravesical instillation with BGC	Acknowledge the limitations of EORTC, suggest combining clinical data and emerging molecular data
BIOMARKERS	Not considered	Promotes research on biomarkers to improve risk stratification, but not yet standardised in clinical practice
BCG THERAPY GUIDANCE	Not completely integrated into risk models	Risk-adapted use of BCG including for intermediate and high-risk patients
CIS MANAGEMENT	Treated as high risk; included in scoring	Lay emphasis on aggressive management due to high risk of progression
FOLLOW-UP RECOMMENDATIONS	Based on risk group; Frequent cystoscopies and cytology	More individualised follow-up based on updated risk groups and BCG status

**Table 2 cancers-17-03078-t002:** Based on ClinicalTrials.gov online database from the National Library of Medicine (NIH).

MIBC	Status of Clinical Trial	Number of Clinical Trial	Completed with Results
	RECRUITING	NCT05241340	NO
	RECRUITING	NCT06305767	NO
	ACTIVE, NOT RECRUITING	NCT02447549	NO
**MET**	COMPLETED	NCT00829920	NO
	COMPLETED	NCT03702179	YES
	ACTIVE, NOT RECRUITING	NCT02546661	NO
	COMPLETED	NCT04209114	YES
	RECRUITING	NCT05544552	NO
	COMPLETED	NCT01031420	YES
**FGFR**	COMPLETED	NCT02177695	YES
	ACTIVE, NOT RECRUITING	NCT03775265	NO
	COMPLETED	NCT00380029	YES
	RECRUITING	NCT06511648	NO
	RECRUITING	NCT05316155	NO

Growth factor receptors: MET, hepatocyte growth factor receptor; FGFR, fibroblast growth factor receptor. Muscle invasive bladder cancer (MIBC).

**Table 3 cancers-17-03078-t003:** Frequency in NOTCH mutation in Non-Muscle-Invasive Bladder Cancer (NMIBC), Muscle-Invasive Bladder Cancer (MIBC) and Overall Bladder Cancer (Overall BC) (cBioPortal Data, https://www.cbioportal.org/, accessed on 16 June 2025).

Frequency of Mutated Genes	NMIBC	MIBC	Overall BC
NOTCH 1	2.9%	3.4%	3.8%
NOTCH 2	1.0%	3.5%	3.7%
NOTCH 3	2.9%	4.4%	4.8%
NOTCH 4	4.8%	4.0%	4.3%
NUMBER OF PROFILED SAMPLES	105	1200	5265

**Table 4 cancers-17-03078-t004:** Predictive analysis of the functional impact of genetic variants.

Protein Change	Type	Consequences	VEP Impact	SIFT Impact	PolyPhen Impact
R169C	Substitution	Missense	MO	DH	PO
W1750*	Substitution	Stop Gained	HI	--	--
H1690Y	Substitution	Missense	MO	DH	BE
E1585K	Substitution	Missense	MO	DH	PR
R1568Q	Substitution	Missense	MO	DH	BE
1138Sfs*132	Deletion	Frameshift	HI	--	--
S834N	Substitution	Missense	MO	TO	BE
S1688C	Substitution	Missense	MO	TO	BE
S1542L	Substitution	Missense	MO	DH	BE
E538K	Substitution	Missense	MO	TO	BE
Q214H	Substitution	Missense	MO	DH	PR
L1654=	Substitution	Synonymous	LO	--	--
A1796S	Substitution	Missense	MO	DH	PO
V633=	Substitution	Synonymous	LO	--	--
E1161K	Substitution	Missense	MO	DH	PO
R2009W	Substitution	Missense	MO	DH	PR
D1587N	Substitution	Missense	MO	DH	PR
E309K	Substitution	Missense	MO	TO	PO
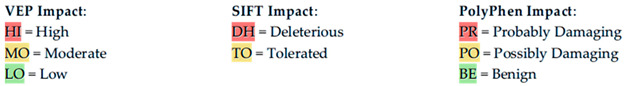

This table presents 18 somatic mutations for NOTCH3 protein, derived from 771 cases analysed by the TCGA platform (https://www.cancer.gov/ccg/research/genome-sequencing/tcga, accessed on 17 June 2025). It is a comprehensive analysis of various protein-coding variants, detailing their type, molecular consequences, and predicted functional impact according to three in silico tools: VEP (Variant Effect Predictor), SIFT, and PolyPhen. The “Protein Change” column lists the amino acid substitutions identified. The “Type” column specifies whether the variant is a substitution or a deletion, while the “Consequences” column describes the predicted effect at the protein level, such as missense, synonymous, or stop gained. The symbol “*” indicates a premature stop codon, meaning the protein translation ends earlier than expected. The symbol “=” indicates a synonymous change: the DNA variant does not alter the amino acid sequence.

## Data Availability

The data that supports the findings of this study are available on request from the corresponding author and F.S.

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
