# Peer review of "Bladder Cancer: Uncovering the Predictive Role of NOTCH as an Emerging Candidate Biomarker for Therapeutic Strategies"

_cancers, 2025, doi:10.3390/cancers17183078_

Round 1
Reviewer 1 Report
Comments and Suggestions for Authors
While the manuscript touches on a meaningful and potentially impactful subject, it currently falls short of the standards expected for a high-quality scientific review. Major revisions are necessary, particularly in terms of content depth, scientific reasoning, and writing quality, to improve its scholarly value and suitability for publication.
- The role of NOTCH signaling (especially NOTCH2/3) in BCa has been previously studied and reviewed, as the manuscript itself admits. The review does not add clear new insights or hypotheses.
- It fails to provide novel mechanistic interpretation, functional models, or propose new interactions—for instance, how NOTCH2 uniquely interacts with tumor microenvironment, immune escape, or drug resistance mechanisms.
- No integration with current omics or AI approaches, which could make the biomarker discovery process more robust and forward-looking.
- The abstract begins mid-sentence and lacks grammatical completeness (“cancer (BCa) is the tenth…”). In addition, the core contents of the introduction have not been clearly presented. For example, 1) what is the existing question? 2) what is your solution? 3) what is the expected outcome, and what is the potential impact of this study? Much of the section on clinical classifications, therapies, and risk scoring repeats standard textbook knowledge not directly tied back to the central NOTCH2 biomarker topic.
- The manuscript does not sufficiently differentiate between the roles of NOTCH1, 2, and 3 with supporting literature or a mechanistic model. In addition, the relationship between NOTCH signaling and therapy resistance, immune modulation, or epithelial–mesenchymal transition (EMT) in BCa is not explored.
Author Response
Dear Editor and Referees,
We appreciate the time and effort you invested in our manuscript. We replied to the Referee's comments and suggestions with the aim of reshaping the text to improve the content and to increase the readability. We carefully read the criticism by the Referees, and we hope we fulfilled all the points. We highlighted in yellow the new parts and added one new figure and one new table.
Reviewer 1#
Comments and Suggestions for Authors
While the manuscript touches on a meaningful and potentially impactful subject, it currently falls short of the standards expected for a high-quality scientific review. Major revisions are necessary, particularly in terms of content depth, scientific reasoning, and writing quality, to improve its scholarly value and suitability for publication.
- The role of NOTCH signaling (especially NOTCH2/3) in BCa has been previously studied and reviewed, as the manuscript itself admits. The review does not add clear new insights or hypotheses.
We thank the reviewer for the accurate revision of our manuscript and the interesting clues to discuss in the manuscript. In this review, we focused on summarising the findings on NOTCH in bladder cancer obtained to date, but particularly addressing the new data of the most recent ones about the oncogenic role of NOTCH2 and NOTCH3. The role of the three NOTCH receptors is ambiguous in this tumoral context as compared to T-cell leukaemia, where the oncogenic potential of NOTCH, particularly NOTCH1 and NOTCH3, has been well established (PMID: 21217079; 36674902, 39684550, and more). Based on the past and recent reports on Bladder Cancer (BCa), data converge in proposing that both NOTCH2 and NOTCH3 have oncogenic potential to promote BCa development and its effect on the tumour microenvironment and response to drugs. Still to be clarified is the precise role of NOTCH4, of which we discussed the very recent reports, but additional studies are required. In Table 3, we used cBioPortal data to analyze NOTCH gene mutations comparatively in NMIBC, MIBC, and overall Bladder cancer. According to us, NOTCH3 should be considered an interesting oncogenic element in this tumour, but has only recently been highlighted. Therefore, we dedicated our attention to implementing NOTCH3 information by including in the manuscript our analysis (Figures 2 and 3; Tables 3 and 4) about mutations and co-lesions essentially in MIBC where NOTCH3 is particularly involved. Hoping to introduce the reader in this topic, we focused on analysing the recurrence of NOTCH3 mutation in both NIMBC and MIBC and the driver mutations of the receptor, all of which are new data never published so far. Based on your suggestion and our hypothesis about NOTCH3, we do believe that many efforts are required to unravel the role of the Notch family in BCa. We modulate the text to enhance the suggestion by the Reviewer. In the final part of the review, where the Conclusions and Future Perspectives are reported, we added some comments and suggestions for subsequent scientific studies and some hypotheses on the therapies that seem to be most promising.
- It fails to provide novel mechanistic interpretation, functional models, or propose new interactions—for instance, how NOTCH2 uniquely interacts with tumor microenvironment, immune escape, or drug resistance mechanisms.
Regarding the Reviewer's concern, we modulated the text and added new data recently published about NOTCH2.
On page 14 of 22, we already discussed: “As reported by Bin Y's research, NOTCH 3, not only NOTCH 2, seems to interact with the tumour microenvironment but also cause stronger immune escape and thus a weaker ability to respond to immunotherapy in bladder cancer.” This could be explained by the fact that both NOTCH2 and NOTCH3 play a role as oncogenes in BCa, unlike NOTCH1, which is a tumour suppressor.
We added a new paragraph to the text on page 11 of 22.
“Interestingly, two recent manuscripts evidenced that NOTCH2 gene alterations correlate with immune infiltration and response to therapy in Bladder Cancer (PMID: 40016843; PMID: 39755215). Indeed, the study by Nagumo Y et al. (PMID: 39755215), by integrating gene alterations and tumor-infiltrating lymphocyte profiling, could associate NOTCH2 mutations with clinical complete response to therapy (atezolizumab and radiation therapy) in MIBC. On the contrary, FGFR3 alterations are associated with non-complete response to immune-radiation therapy, in line with its association with aggressive BCa. The different therapy response was related to the CD8:Foxp3 lymphocyte ratio, higher in complete response to therapy, thus interestingly correlating NOTCH2 to the composition of the tumor microenvironment. The Authors hypothesize that NOTCH2, in association with CDK12, GNAS, and AR1D1A gene alterations, could identify good responders to therapy in MIBC. To support the notion that NOTCH2 behaves as an oncogene is the manuscript by Si-yu Chen et al (PMID: 40016843). Besides the role of NOTCH2 in promoting cell proliferation and metastasis through EMT, this receptor is involved in BCa stemness, a crucial step in disease progression.”
All the evidence converges on the oncogenic role of both NOTCH2 and NOTCH3. Does it depend on the different receptor structure or cell context? As suggested by the Reviewer, we are convinced that the topic is “meaningful and potentially impactful” to understand the disease, and multiple data consider NOTCH receptors as reliable markers, a hypothesis that we favour. Therefore, many antibodies raised against NOTCH could be suitable drugs for future tailored therapy to integrate the standard ones.
- No integration with current omics or AI approaches, which could make the biomarker discovery process more robust and forward-looking.
We thank the Reviewer for the comment and have added a paragraph in Conclusions and Future Perspectives. Recent advances integrating multi-omics and AI in bladder cancer have enabled the identification of fibroblast-related biomarkers and a prognostic index for patient stratification (PMID: 38759695), as well as the development of a T cell–specific five-gene signature (TstcSig) capable of accurately predicting immunotherapy response (PMID: 39033098). Together, these studies demonstrate how multi-layered molecular profiling can both dissect the complexity of the tumor microenvironment and generate clinically actionable biomarkers. Building on this evidence, similar strategies could be applied to the study of Notch signaling, with the potential to define novel molecular subtypes and guide the development of tailored combination therapies.
- The abstract begins mid-sentence and lacks grammatical completeness (“cancer (BCa) is the tenth…”). CORRETTO: Bladder cancer (BCa)…
In addition, the core contents of the introduction have not been clearly presented. For example, 1) what is the existing question? 2) what is your solution? 3) what is the expected outcome, and what is the potential impact of this study? Much of the section on clinical classifications, therapies, and risk scoring repeats standard textbook knowledge not directly tied back to the central NOTCH2 biomarker topic.
Based on the Reviewer's suggestion, we modified some parts of the abstract and the initial sentence. In the section on Classification, clinical aspects and therapy, new paragraphs are highlighted in yellow. This section aims to introduce the heterogeneity of Bladder Cancer, which may reflect the multiple mutations contributing to disease development and resistance to therapy
- The manuscript does not sufficiently differentiate between the roles of NOTCH1, 2, and 3 with supporting literature or a mechanistic model.
We agree with the Reviewer that it is necessary to distinctly discuss the four receptors. In the manuscript, we analyse NOTCH1, 2, 3, and 4 in separate paragraphs to precisely highlight their structural and functional differences in BCa. As can be seen from the review, based on scientific literature, NOTCH1 plays a tumour-suppressing role, unlike NOTCH2 and 3, which instead play an oncogenic role. More complicated and underexplored is the role of NOTCH4. Moreover, for NOTCH3, we newly report mutations, in terms of localization and percentages, in MIBC, in which this receptor is particularly involved. In agreement with published reports, we favour the hypothesis that the different NOTCH receptors might distinguish and identify distinct stages of the BCa, thus suggesting their role as reliable markers of the disease. Detection of a “good biomarker” could guide drug choice for a timely and efficacious therapy, as well as for a more intense follow-up. In reply to the Reviewer's concern and to ameliorate the text, we remodelled the related paragraphs.
In addition, the relationship between NOTCH signaling and therapy resistance, immune modulation, or epithelial–mesenchymal transition (EMT) in BCa is not explored.
We thank the Reviewer for the suggestion and have modified some parts accordingly. We added in the NOTCH2 section a new paragraph in yellow, already included in the previous Reviewer reply at point 2. Nevertheless, the manuscript contains related information. On page 11 of 22. Lin H.'s study (PMID: 38462037) observes the role of the LRP1/DLL4/NOTCH2 axis in promoting EMT progression and hypothesizes a role in resistance to immunotherapy. Furthermore, Notch signaling has been implicated in driving epithelial–mesenchymal transition (EMT), thereby promoting tumor progression and metastasis in cancer (PMID: 23696246). Additionally, on lines ... refers to the study by Bin Y. (PMID: 38558562), which identified 10 genes related to the NOTCH signalling pathway and divided bladder cancer into two subtypes (groups C1 and C2) based on the expression of these genes. It was observed that these 10 genes are closely related to prognosis, showing a high risk for group C1 (with upregulation of ADAM17, DTX3L, MAML2, SNW1, NOTCH3, NUMBL). They also compared the differences between common cells and the scores relating to the immune system of the tumour microenvironment (TME), discovering that group C1 had a greater ability to evade the immune system and a lower ability to respond to immunotherapy. Moreover, the Notch pathway, modulated by SCAMP2, contributes to cisplatin resistance in bladder cancer by promoting tumor cell survival under chemotherapeutic stress (PMID: 40406117). Targeting SCAMP2 or Notch signaling, therefore, emerges as a potential strategy to overcome chemoresistance and enable more personalized therapeutic approaches.
Reviewer 2 Report
Comments and Suggestions for Authors
The author have done a review of different biomarkers in BC with most focus on the NOTCH genes. It is a well written manuscript.
Minor comment: Page 3, line 95: "At the state" I do not understand what the authors mean.
Author Response
Dear Editor and Referees,
We appreciate the time and effort you invested in our manuscript. We replied to the Referee's comments and suggestions with the aim of reshaping the text to improve the content and to increase the readability. We carefully read the criticism by the Referees, and we hope we fulfilled all the points. We highlighted in yellow the new parts and added one new figure and one new table.
Reviewer 2#
Comments and Suggestions for Authors
The author have done a review of different biomarkers in BC with most focus on the NOTCH genes. It is a well written manuscript.
Minor comment: Page 3, line 95: "At the state" I do not understand what the authors mean.
We thank the Reviewer for the positive and encouraging comments.
In reply to his/her suggestion and to improve the readability of the text, we removed the sentence.
Reviewer 3 Report
Comments and Suggestions for Authors
BLADDER CANCER: uncovering the predictive role of NOTCH as an emerging candidate biomarker for therapeutic strategies, is a comprehensive review on bladder cancer.
The review can benefit from addition of the expression levels of NOTCH and related pathways in bladder cancer.
A table of ongoing clinical trials in Bladder cancer and if there are studies showing alterations in the notch upon drug treatment.
There is mention of immunotherapy, again how is the notch expression alterated under treatment.
Gene has to be in capitals across the manuscript.
This statement "In support, a histone deacetylase (HDAC) inhibitor, suberoylanilide hydroxamic acid 410 (SAHA), increases NOTCH3 protein acetylation levels by decreasing NOTCH3 expres- 411 sion, thereby preventing urothelial cancer cell proliferation" is not clear.
Author Response
Dear Editor and Referees,
We appreciate the time and effort you invested in our manuscript. We replied to the Referee's comments and suggestions with the aim of reshaping the text to improve the content and to increase the readability. We carefully read the criticism by the Referees, and we hope we fulfilled all the points. We highlighted in yellow the new parts and added one new figure and one new table.
Reviewer 3#
Comments and Suggestions for Authors
BLADDER CANCER: uncovering the predictive role of NOTCH as an emerging candidate biomarker for therapeutic strategies, is a comprehensive review on bladder cancer.
The review can benefit from addition of the expression levels of NOTCH and related pathways in bladder cancer.
In replying to the interesting point of the Reviewer, we remodelled the text of the manuscript to make it clearer and to evidence what was already reported in the manuscript.
A table of ongoing clinical trials in Bladder cancer and if there are studies showing alterations in the notch upon drug treatment.
For the interesting concern by the Reviewer, we prepared Table 2, with selected ongoing clinical trials in muscle-invasive BCa, based on the gene alterations of the most represented growth factor receptors, FGFR3 and MET, both frequently associated with NOTCH3 mutations in MIBC. Unfortunately, no clinical trial involved any NOTCH, but its identification as a reliable marker may include these receptors in the future. We also checked for ATM, PIK3CA, ERCC2, ERBB2, and FBXW7, but no clinical trial involved their gene alterations. FGFR and MET are two of the key growth factor receptors driving bladder cancer development and progression. Their central role in tumor biology has made them prime candidates for therapeutic targeting, which is why ongoing and completed clinical trials have focused on evaluating inhibitors against these pathways.
There is mention of immunotherapy, again how is the notch expression alterated under treatment.
Currently, there are no active clinical trials testing Notch pathway inhibitors—either as individual drugs or in combination—in patients with bladder cancer. The available evidence is limited to promising preclinical studies on NOTCH2, a potential therapeutic target (PMID: 26769750). We would be confident that the identification of NOTCH3 mutations as a frequent gene co-lesion of growth factor receptors (Figure 3) and preclinical data on NOTCH2 may suggest the inclusion of these receptors in future and more in-depth studies.
Gene has to be in capitals across the manuscript.
Thank you to the Reviewer's suggestion, we went through the text and corrected the gene name.
This statement "In support, a histone deacetylase (HDAC) inhibitor, suberoylanilide hydroxamic acid 410 (SAHA), increases NOTCH3 protein acetylation levels by decreasing NOTCH3 expres- 411 sion, thereby preventing urothelial cancer cell proliferation," is not clear.
In reply to Reviewer's concern, we rewrote the sentence, page 13 of 22 as follows: "Increased NOTCH3 protein acetylation by suberoylanilide hydroxamic acid 410 (SAHA), a histone deacetylase (HDAC) inhibitor, decreases NOTCH3 protein levels, which correlate with reduced tumour growth in a xenograft model and with enhanced sensitization of urothelial cancer cells to cisplatinum (PMID: 28416766). Considering the high NOTCH3 expression and poor prognosis demonstrated in urothelial carcinoma (PMID: 28416766), HDAC inhibitors could become an effective therapeutic strategy”.
Reviewer 4 Report
Comments and Suggestions for Authors
Major Comments
In the Introduction, it would be helpful to include a classification of bladder cancer (BCa) and relate it to mortality rates as well as the recurrence and mortality rates of each subtype.
In Figure 2, the use of colors alone to distinguish the groups may cause difficulty in interpretation. Please add numbers to indicate each group and include the n value for clarity.
An overview figure of the NOTCH1–3 pathway would considerably enhance the clarity of the manuscript and assist readers in interpreting the results. The inclusion of such a figure is therefore recommended.
Minor Comments
In Table 2, the frequencies of NOTCH1–3 mutations are presented, but the classification is labeled only as “Bladder cancer.” Revise the label to “Overall bladder cancer”
The abbreviations used in the titles of Sections 4 and 5 are inconsistent. Please revise them for uniformity, for example by using “NMIBC” for Non-muscle-invasive bladder cancer and ensuring consistent use of “MIBC.”
Comments on the Quality of English Language
Not applicable
Author Response
Dear Editor and Referees,
We appreciate the time and effort you invested in our manuscript. We replied to the Referee's comments and suggestions with the aim of reshaping the text to improve the content and to increase the readability. We carefully read the criticism by the Referees, and we hope we fulfilled all the points. We highlighted in yellow the new parts and added one new figure and one new table.
Reviewer 4#
Comments and Suggestions for Authors
Major Comments
In the Introduction, it would be helpful to include a classification of bladder cancer (BCa) and relate it to mortality rates as well as the recurrence and mortality rates of each subtype.
In reply to the motivated Reviewer concerns, we added the information requested, highlighted in yellow within the text of the manuscript, inside the section Classification, clinical aspects and therapy: on page 3 of 22; on page 4 of 22.
In Figure 2, the use of colors alone to distinguish the groups may cause difficulty in interpretation. Please add numbers to indicate each group and include the n value for clarity.
We thank the Reviewer, and to improve the figure readability, we number each section of the pie chart and describe the analysed gene alteration/s and the corresponding number of case/s in the associated legend. We believe that the remodelling of the figure will provide clearer information to the reader. Figure 2 is now numbered as Figure 3.
An overview figure of the NOTCH1–3 pathway would considerably enhance the clarity of the manuscript and assist readers in interpreting the results. The inclusion of such a figure is therefore recommended.
To reply to the Reviewer's suggestion and to help the reader across the manuscript, we prepared and added a new figure, Figure 1, highlighting the main steps in NOTCH signaling.
Minor Comments
In Table 2, the frequencies of NOTCH1–3 mutations are presented, but the classification is labeled only as “Bladder cancer.” Revise the label to “Overall bladder cancer”
As requested by the Reviewer, we modified Table 2 by replacing “Overall Bladder Cancer”.
Round 2
Reviewer 3 Report
Comments and Suggestions for Authors
no comments